# Prioritization of Critical Factors for Surveillance of the Dissemination of Antibiotic Resistance in *Pseudomonas aeruginosa*: A Systematic Review

**DOI:** 10.3390/ijms242015209

**Published:** 2023-10-15

**Authors:** Jung Hun Lee, Nam-Hoon Kim, Kyung-Min Jang, Hyeonku Jin, Kyoungmin Shin, Byeong Chul Jeong, Dae-Wi Kim, Sang Hee Lee

**Affiliations:** 1National Leading Research Laboratory of Drug Resistance Proteomics, Department of Biological Sciences, Myongji University, Yongin 17058, Republic of Korea; junghunlee@mju.ac.kr (J.H.L.); cardgame79@gmail.com (K.-M.J.); ehdqhdl1@naver.com (H.J.); hiu980305@naver.com (K.S.); bcjeong@mju.ac.kr (B.C.J.); 2Division of Life Sciences, Jeonbuk National University, Jeonju 54896, Republic of Korea; krkimdoller@jbnu.ac.kr

**Keywords:** *Pseudomonas aeruginosa*, antibiotic resistance genes, mobile genetic elements, sequence types, One Health

## Abstract

*Pseudomonas aeruginosa* is the primary opportunistic human pathogen responsible for a range of acute and chronic infections; it poses a significant threat to immunocompromised patients and is the leading cause of morbidity and mortality for nosocomial infections. Its high resistance to a diverse array of antimicrobial agents presents an urgent health concern. Among the mechanisms contributing to resistance in *P. aeruginosa*, the horizontal acquisition of antibiotic resistance genes (ARGs) via mobile genetic elements (MGEs) has gained recognition as a substantial concern in clinical settings, thus indicating that a comprehensive understanding of ARG dissemination within the species is strongly required for surveillance. Here, two approaches, including a systematic literature analysis and a genome database survey, were employed to gain insights into ARG dissemination. The genome database enabled scrutinizing of all the available sequence information and various attributes of *P. aeruginosa* isolates, thus providing an extensive understanding of ARG dissemination within the species. By integrating both approaches, with a primary focus on the genome database survey, mobile ARGs that were linked or correlated with MGEs, important sequence types (STs) carrying diverse ARGs, and MGEs responsible for ARG dissemination were identified as critical factors requiring strict surveillance. Although human isolates play a primary role in dissemination, the importance of animal and environmental isolates has also been suggested. In this study, 25 critical mobile ARGs, 45 critical STs, and associated MGEs involved in ARG dissemination within the species, are suggested as critical factors. Surveillance and management of these prioritized factors across the One Health sectors are essential to mitigate the emergence of multidrug-resistant (MDR) and extensively resistant (XDR) *P. aeruginosa* in clinical settings.

## 1. Introduction

*Pseudomonas aeruginosa* is a Gram-negative bacterium belonging to the class Gammaproteobacteria that can survive and thrive in diverse environments as a ubiquitous environmental bacterium [1]. It is a primary opportunistic human pathogen that is responsible for various acute infections, and poses a substantial threat to immunocompromised patients as the leading cause of morbidity and mortality in nosocomial infections [2,3]. The bacterium’s versatile metabolic ability enables it to outcompete other bacteria in nutrient-limited conditions, suggesting its adaptability to hospital environments, and leading to nosocomial infections [4]. The prevalence of *P. aeruginosa* in healthcare-associated infections underscores the significance of the bacterium as an opportunistic pathogen [5]. *P. aeruginosa* plays a prominent role as a major causative agent of chronic respiratory infections, as exemplified by its involvement in infections in patients with cystic fibrosis (CF) and chronic obstructive pulmonary disease [6,7]. Moreover, this bacterium has been implicated in a wide spectrum of infections, including community-acquired pneumonia, urinary tract infection, wound infection, and bacteremia [8]. Considering the wide range of infections caused by *P. aeruginosa* and their infection frequencies, a range of chemical regimens have been implemented in the clinical settings as empirical antibiotic therapies aimed at mitigating these infections [9,10]. However, the resistance profiles of *P. aeruginosa* to a broad spectrum of antibiotics coupled with the misuse and overuse of antibiotics, have led to the emergence of multidrug-resistant (MDR) and extensively resistant (XDR) *P. aeruginosa*, constituting a substantial concern in clinical settings [11].

*P. aeruginosa* is a constituent of the “ESKAPE’ pathogens (*Enterococcus faecium*, *Staphlylococcus aureus*, *Klebsiella pneumoniae*, *Acinetobacter baumannii*, *P. aeruginosa*, and *Enterobacter* species), that elicit clinical concern due to their abilities to cause life-threatening nosocomial infections and their broad-spectrum antibiotic resistance profiles [12,13]. Furthermore, according to the WHO priority list of antibiotic-resistant bacteria necessitating research and development of effective drugs, *P. aeruginosa* has been enrolled in the “critical” tier (priority 1 group among critical, high, and medium priorities) [14]. This emphasizes the great clinical concern posed by the antibiotic resistance of *P. aeruginosa* and highlights the imperative need for a comprehensive understanding of resistance to effectively mitigate the emergence of MDR and XDR *P. aeruginosa*. The elevated and extensive resistance profiles exhibited by *P. aeruginosa* were attributed to the interplay among intrinsic, adaptive, and acquired resistance mechanisms. Intrinsic resistance involved a number of factors, including lower outer membrane permeability, the presence of an antibiotic efflux pump, and the presence of OXA-50 (oxacillinase 50) and PDC (*Pseudomonas*-derived cephalosporinase) β-lactamases [15]. Adaptive resistance is characterized by resistance that emerges in response to the persistent presence of antibiotics or other environmental stimuli and is exemplified by the inducible expression of PDC β-lactamases and MexXY efflux pumps [16,17]. Acquired resistance mechanisms include the horizontal acquisition of antibiotic resistance genes (ARGs) and mutational resistance. The horizontal acquisition of ARGs is facilitated by a diverse array of mobile genetic elements (MGEs), such as plasmids, transposons, integrons, insertion sequences (ISs), prophages, genomic islands (GIs), and integrative conjugation elements (ICEs) [18]. Mutational resistance refers to the capacity for resistance development through the selection of chromosomal mutations [6,19]. The significance of mutational resistance is particularly pronounced in chronic infections such as lung infections in patients with CF due to the prevalence of hypermutable strains [10]. The relatively large genome size of *P. aeruginosa* and its genetic plasticity are responsible not only for its versatile metabolic adaptability to diverse environments, but also for facilitation of the acquisition of antibiotic resistance [20,21,22]. Although both acquired resistance mechanisms contribute to resistance in MDR/XDR *P. aeruginosa*, their emergence harboring horizontally transferred ARGs is increasingly perceived as a burgeoning menace in clinical settings [6,23]. Numerous studies have been dedicated to elucidating the horizontally acquired ARGs that exhibit clinically important resistance such as extended-spectrum β-lactamase and carbapenemase, within high-risk sequence type (ST) clones [6,23,24,25]. The primary factors restricting therapeutic options and leading to unexpected clinical outcomes in *P. aeruginosa* infections are the virulence and antibiotic resistance of high-risk ST clones [24]. Therefore, a comprehensive understanding of the correlation between STs and horizontally acquired ARGs in *P. aeruginosa* is required. Moreover, there is a growing consensus that understanding the infectivity and antibiotic resistance of *P. aeruginosa* should adopt a One Health perspective considering its colonization within diverse environmental ecosystems [26]. It is crucial to gain a comprehensive understanding of the flow of the horizontally acquired ARGs among isolates across One Health sectors (human, animal, and environment) given the ubiquitous nature of *P. aeruginosa* in diverse ecological niches [1,26,27].

The primary objective of this study was to compile a comprehensive list of horizontally acquired ARGs in *P. aeruginosa* with a specific focus on characterizing mobile ARGs associated with MGEs. Additionally, we assessed the prevalence of these ARGs among the One Health sectors and STs to identify the critical sectors and STs contributing to the dissemination of ARGs in *P. aeruginosa*. We adopted two systematic approaches to achieve these goals. First, a systematic literature analysis was performed using the Preferred Reporting Items for Systematic Reviews and Meta-Analyses (PRISMA) strategy [28]. Although this strategy is invaluable for obtaining results that are related to well-documented topics, it is limited to the information provided by the authors and may overlook unanalyzed data. To complement this approach, we conducted a genome database survey using public genome databases. This approach allowed us to scrutinize all the available sequence information and various attributes of the isolates, thus providing an overall understanding of *P. aeruginosa* ARGs, MGEs, and pathogenic STs from a One Health perspective. By combining both approaches, we aimed to consolidate a critical inventory of mobile ARGs, MGEs, and STs in *P. aeruginosa* that should be under strict surveillance.

## 2. Methods

### 2.1. Systematic Literature Analysis

The PRISMA guidelines were employed for the systematic analyses of ARG dissemination via MGEs [28], and a recently published article was referred to for detailed procedures [29]. We conducted comprehensive searches in scientific literature databases including PubMed, MEDLINE, and Embase by utilizing a combination of three keywords from each of the following categories: category 1, *Pseudomonas aeruginosa*; category 2, antibiotic resistance, antimicrobial resistance, specific class or antibiotic resistance, and ARG(s); category 3, dissemination, transmission, mobile genetic element(s), MGE(s), horizontal gene transfer, HGT, plasmid(s), transposon(s), integron(s), and insertion sequence(s). The searches were confined to title and abstract and we focused on scientific literature that was published between 2000 and 2023 in English. Our analyses were designed to thoroughly investigate ARGs in *P. aeruginosa* and derive their associations with MGEs to thereby reveal the dissemination of ARGs within this opportunistic pathogen via MGEs. Initially, 883 non-redundant publications were identified. Among these, we excluded 572 records that included review articles, articles without full-text, articles devoid of ARGs data, and articles without useful data (limited to PCR detection of specific ARGs or phenotypic analyses), ultimately resulting in 311 studies. Subsequently, 129 articles that did not establish a clear association between ARGs and MGEs were excluded. Consequently, a final subset of 182 studies was identified for the systematic review with the aim to enhance our understanding of ARG dissemination via MGEs in *P. aeruginosa* (Figure 1).

### 2.2. Genome Database Survey

We conducted a thorough investigation of ARGs in *P. aeruginosa* genomes and their associations with MGEs using the NCBI RefSeq genome database. Our approach aligns with the growing emphasis on the genome-based prediction and surveillance of ARGs [30]. High-throughput comparative genomics can provide insights into the phylogeny, evolution, and horizontal gene transfer of ARGs and virulence genes in pathogenic bacteria [31]. We downloaded 9774 RefSeq *P. aeruginosa* genome assembly sequences from the NCBI datasets (https://www.ncbi.nlm.nih.gov/datasets/, accessed on 10 May 2023). The strategy for genome database analyses comprises two different approaches that include a sequence-based search and an annotation-based search (Figure 2).

In the sequence-based search, we utilized genome assemblies as queries for the Resistance Gene Identifier (RGI) and PubMLST pipelines to perform ARG searches and multilocus sequence typing of genomes (MLST), respectively (Figure 2 and Appendix A) [32,33]. For ARG searches, we used the Comprehensive Antibiotic Resistance Database (ver. 3.2.7) as of May 2023. ARGs in the database were grouped using sequence identity-based clustering with a 90% identity cut-off, ultimately resulting in representative ARGs for each group (Appendix A). Mutational resistance profiles were not analyzed, as our study focused on the horizontal transfer and further dissemination of ARGs mediated by MGEs. To account for comparisons among groups with varying numbers of genomes in the database, generally we used the ARG copy number per genome or proportion in genomes.

The annotation-based search was employed to gather information on the metadata and MGEs of the isolates (Figure 2). For metadata collection, we utilized the Biosample database linked to the NCBI RefSeq genome data. We categorized the attributes of the isolates based on their origins (human, animal, or environment) (Appendix A). For the human isolates, additional human specimen information was collected (Appendix A). We searched for integrase and transposase proteins in the genomes with protein annotations (7995 genomes) as markers of integrons and transposons, respectively (Appendix A).

Statistical significance among the abundance of acquired ARGs in One Health sectors was evaluated using Student *t*-test. The Bray-Curtis dissimilarity matrix was employed to cluster STs based on the relative proportion of ARGs repertoires in all genomes belonging to each ST. The correlation between the relative abundances ARGs and MGEs was analyzed using the Pearson correlation analysis to define the positive correlation (r > 0.1) and statistical significance (*p* < 0.001).

## 3. Results

### 3.1. Systematic Literature Analysis

#### 3.1.1. MGEs Linked to ARGs in *P. aeruginosa*

To ascertain the prevalence of major MGEs associated with ARGs, we surveyed the frequency of studies involving any of six primary types of MGEs, including integrons, plasmids, transposons, ISs, ICEs, and GIs (Figure 3A and Appendix A). Among the 182 studies that were examined, cases of ARG-carrying integrons were observed in 105 (57.3%) studies. With three exceptions, all detected integrons were defined as class 1 integrons (Appendix A), and this was consistent with observations in many other Gram-negative pathogenic bacteria [34]. Plasmids were the second most frequent ARG-carrying MGEs (55 studies) (Appendix A). Transposons and ISs associated with ARGs were identified in 34 and 28 studies, respectively (Appendix A). While there were a small number of studies addressing ICE and GI in relation to ARGs, their contribution to ARG dissemination appeared to be relatively minor based on their low frequency of occurrence (less than 10 studies each) (Figure 3A). A total of 41 studies indicated the presence of greater than two MGEs linked to ARGs (Figure 3B). Notably, class 1 integrons were closely associated with the plasmids, thus indicating the presence of class 1 integrons on the plasmids (Figure 3B). The frequencies of these types were investigated to identify the critical ISs and transposons responsible for ARG dissemination. Among the transposons, Tn*3* was the most frequently reported ARG-carrying transposon, and this was followed by Tn*402*, Tn*4401b*, Tn*1403*, Tn*4371*, and Tn*6346* (Figure 3C). Details regarding passenger ARGs within these transposons are presented in Appendix A. Regarding IS elements, no prominent ISs harboring ARGs were identified, although IS*26* carrying *bla*_KPC-2_ and *qnrS1* was the most frequently reported IS (Figure 3C and Appendix A).

#### 3.1.2. Major STs Involved in ARG Dissemination via MGEs

Although not all studies included in the analysis provided ST information for *P. aeruginosa*, 56 studies indicated the ST of the pathogenic bacterium. This allowed us to compile data detailing the prevalence of the STs associated with ARG dissemination. A recent study suggested the top 10 nosocomial MDR/XDR *P. aeruginosa* high-risk clones based on their prevalence, global distribution, and MDR/XDR profiles with a specific focus on horizontally acquired β-lactamases [23]. The top 10 clones (ST235, ST111, ST233, ST244, ST357, ST308, ST175, ST277, ST654, and ST298) were the most extensively studied clones in 182 articles, thus indicating that they are important clones for ARG dissemination via MGEs (red-colored STs in Figure 3D). Despite not being included in the top 10 list, the positions of ST463 as the 7th most abundant clone highlights its prevalence in the studies. ST463 has recently gained attention as a potentially high-risk clone due to its virulence and antibiotic resistance [24].

#### 3.1.3. Mobile ARGs

Elucidating mobile ARGs is of paramount importance for the surveillance of the flow and emergence of ARGs in *P. aeruginosa* [27]. Mobile ARGs are defined as ARGs exhibiting mobile traits or linkages to MGEs. Through our literature survey, we identified mobile ARGs carried by integrons, plasmids, transposons, and ISs (Figure 3E and Appendix A). To conduct accurate frequency analyses of these ARGS, we unified ARG names as representative ARG names based on a 90% identity cut-off for clustered ARGs from the Comprehensive Antibiotic Resistance Database (CARD) (Appendix A). Integron-borne ARGs were dominant and among those, sulfonamide resistance gene (*sul1*) and the disinfectant resistance gene (*qacEΔ1*, MDR pump) that are constituents of the typical structure of clinical class 1 integron, exhibited the most frequent appearance in integrons. Passenger ARGs present within integrons included *aac(6′)-Ib, ant(2′’)-Ia, ant(3′’)-IIa, acc(6′)-Il,* and others for aminoglycoside resistance, and *bla*_VIM-1_, *bla*_OXA-10_, and others for β-lactam resistance (Figure 3E). ARGs located in plasmids and transposons exhibited patterns similar to those of integrons, thus indicating that some integrons were integrated within the plasmid or transposon structures (Figure 3E). Notably, *bla*_KPC-2_ is a unique ARG observed exclusively in plasmids and transposons without linkages to integrons (Figure 3E). In contrast, the IS-borne ARGs were less abundant. An important observation was the inactivation of the *oprD* gene that was mediated by various IS elements (Figure 3E and Appendix A). Inactivation of *oprD* that encodes a porin responsible for antibiotic uptake (particularly for carbapenem antibiotics) has been reported to be involved in the increase in resistance levels since its first discovery in the context of IS-mediated inactivation [36,37]. Red ARGs presented in Figure 3E represent ARGs that were identified as mobile ARGs in our genome database survey, which were almost prevalent in the literature analysis.

### 3.2. Genome Database Survey

#### 3.2.1. Intrinsic and Acquired ARGs in *P. aeruginosa*

Numerous studies have prioritized ARGs according to their health risks while considering factors such as their origin as human pathogens and their mobility [38,39]. To understand the dissemination of ARGs in *P. aeruginosa*, it is crucial to differentiate between intrinsic and acquired resistance (in this study, we excluded mutational resistance but encompassed horizontally acquired resistance). Although it is important to comprehend the intrinsic resistance at the species level and to develop strategies to control it, the paramount focus lies in identifying the acquired ARGs that are introduced into the species and circulate within it. These acquired ARGs can significantly influence the outcomes of infections following exposure to identical chemical regimens [15].

As a first step toward gaining insights into ARG dissemination within and among *P. aeruginosa* populations, we calculated the copy number of ARGs per genome assembly. Of the 426 representative ARGs that were detected in all the genome assemblies, 44 were categorized as intrinsic ARGs with a copy number per genome of greater than 0.9. This suggests that these ARGs are genetically conserved within this taxon and can be considered predictable elements contributing to the resistance background. In contrast, the copy numbers per genome observed for acquired ARGs were less than 0.4 and most of them fell below 0.2. This clear distinction in copy number per genome serves as a criterion for differentiating between intrinsic and acquired ARGs (Appendix A). A total of 382 ARGs were designated as acquired ARGs, and these were used in subsequent analyses to gain a deeper understanding of ARG dissemination (Appendix A). While these ARGs were certainly acquired, their significance should be further examined by analyzing their correlation with One Health origins, STs, and MGEs.

#### 3.2.2. Acquired ARGs in One Health Sectors

Recently, animal and environmental isolates have gained prominence, particularly in the context of ubiquitous bacteria, as they possess the potential to serve as reservoirs and transmission routes of ARGs to high-risk clones [27]. To assess the roles of animal and environmental isolates in ARG dissemination, we compared the distribution of acquired ARGs within the genomes of the human, animal, and environmental isolates. The highest number of acquired ARGs in a single genome was observed in the human isolates (34 ARGs), and this was followed by the animal isolates (23 ARGs) and the environmental isolates (18 ARGs). In terms of average numbers, the human isolates harbored 4.3 acquired ARGs per genome (5099 genomes), whereas the animal and environmental isolates possessed 3.4 ARGs per genome (137 genomes) and 3.3 ARGs per genome (1120 genomes), respectively. Statistical analyses revealed a significant difference in the amount of ARGs in the human isolates compared to that in the animal and environmental isolates (Figure 4A); thus, indicating that in the context of this species, human isolates are more seriously concerned with the management of ARG dissemination. These results imply that human isolates that are more frequently exposed to antibiotics exhibit a greater propensity to acquire ARGs. This observation was further supported by the comparison of ARG abundance according to antibiotic class, where ARGs against generally used anti-psedomonal agents such as β-lactam and aminoglycoside were more abundant in the human isolates (Figure 4B). Notably, the sulfonamide resistance gene (*sul1*) and the disinfectant resistance gene (*qacEΔ1*, MDR pump) possessing the typical structure of the clinical class 1 integron structure [34] that is one of the major MGEs mediating ARG dissemination in *P. aeruginosa* as indicated in the literature analysis (Figure 3 and Figure 4B) were much more abundant in the human isolates. However, it is essential to highlight that certain animal and environmental isolates carried great than 10 acquired ARGs (Figure 4A); thus, emphasizing that the potential contribution of animal and environmental isolates to ARG dissemination should not be underestimated.

#### 3.2.3. ARG Repertoires and the Correlation with One-Health Sector, ST, and MGEs

As demonstrated by the PRISMA, high-risk ST clones were closely associated with ARG dissemination (Figure 3D). The STs of all of the *P. aeruginosa* genomes used in this study were determined through MLST analyses that revealed 1006 STs among 8855 genomes. For statistical analyses, STs with fewer than ten genomes were excluded, ultimately resulting in a dataset of 132 STs consisting of 6904 genomes. The density of a specific ARG within a specific ST was obtained by dividing the sum of the copy numbers of the specific ARG by the total number of genomes belonging to that specific ST [40]. Densities were calculated exclusively for 382 acquired ARGs, as 44 intrinsic ARGs were conserved in all the strains as genetic backgrounds (typically their densities were approximately 1.0; Appendix A). Based on the acquired ARG profiles, 132 STs were clustered using the Bray–Curtis dissimilarity matrix, thus resulting in four distinct clades characterized by acquired ARG repertoires (Figure 5). To highlight the most abundant ARGs, the ARG densities in 6904 genomes are presented as bar graphs at the top of Figure 5. ARGs linked to MGEs identified in the PRISMA are highlighted in red (Figure 5), thus indicating that many of the mobile ARGs that were studied were also dominant in the genome database survey. STs belonging to clade 1 exhibited a broad range of ARGs spanning various antibiotic classes. ARGs were observed to be more abundant and prevalent within clade 1 than they were in the other clades. Noteworthy ARGs in this clade included various aminoglycoside resistance genes (*aac(6′)-Ib*, *ant(3″)-IIa*, *ant(2″)-Ia*, *aph(3″)-Ib*, *aph(3′)-VIa*, and others), *β*-lactamases (*bla*_OXA-570_, *bla*_VIM-1_, *bla*_KPC-2_, *bla*_OXA-2_, *bla*_GES-1_, and others), macrolide-lincosamide-streptogramin B (MLSB) resistance genes (*mphE* and *msrE*), a sulfonamide resistance gene (*sul1*), a diamonpyrimidine resistance gene (*dfrB5*), phenicol resistance genes (*floR*, *cmlA1*, *cmx*, and *catB3*), a tetracycline resistance gene (*tet(A)*), a quinolone resistance gene (*qnrVC1*), a glycopeptide resistance gene (*vanT*), a rifamycin resistance gene (*arr-2*), and MDR efflux pump genes (*adeF* and *qacEΔ1*) (Figure 5). The high prevalence of *sul1* and *qacEΔ1* genes indicates the involvement of class 1 integrons in ARG dissemination in this clade, whereas in the other clades these were not prevalent. All of the top 10 high-risk clones were present in clade 1, thus reinforcing the correlation between the ARG profiles of this clade and the high-risk clones (red arrows in Figure 5). The STs of clades 2, 3, and 4 were much less diverse and prevalent in the ARG profiles than were those of clade 1. The distinct characteristic of clade 2 was the presence of both *bla*_OXA-570_ and *adeF*. Clades 3 and 4 possessed distinctive profiles characterized by the presence of *bla_OXA-570_* and *adeF*, respectively. A gene (*bla*_OXA-570_) encoding OXA-570 was grouped into the subfamily OXA-60-like in the β-lactamase database (BLDB) [41], but a 90% identity cut-off clustering distinguished OXA-60 and OXA-570 (Appendix A). Peptide resistance genes have not been observed to be prevalent in *P. aeruginosa*, although there have been some reports regarding mobile colistin resistance (*mcr*) genes in *P. aeruginosa* since its discovery [42].

In Figure 6, ARG densities are grouped into corresponding antibiotic classes and compared to the isolate information (One Health origins) and the abundance of MGEs. The genomes of the isolates belonging to the STs of clade 1 were actively analyzed and reported in the genome database; most of them originated from humans. Further examination of the human specimen of isolates in clade 1 revealed that they were isolated from various infection sites including the respiratory tract (non-CF), whereas STs in clade 2 were predominantly isolated from the respiratory tract of patients with CF (Figure 6). These findings align with those of previous studies; thus, indicating a high frequency of hypermutable *P. aeruginosa* in CF lung infections where resistance is primarily caused by mutational resistance rather than the horizontal acquisition of ARGs [10,43]. The correlation between the density of acquired ARGs and the abundance of MGEs also suggested that STs in clade 1 were more actively involved in the dissemination of acquired ARGs than were those in the other clades. The proportion of plasmid-borne ARGs and the average copy numbers of integrases and transposases per genome were higher for STs in clade 1 (Figure 6). Remarkably, the proportion of class 1 integron integrases was distinctly higher in the STs in clade 1, and this coincided with the high frequency of sul1 and qacEΔ1 in these STs (Figure 6). Collectively, the distribution of acquired ARGs appears to be dependent upon STs. STs in clade 1 are involved in various human infections and are postulated to be the major carriers and dissemination routes of acquired ARG within this species.

A total of 45 STs belonging to clade 1, which exhibited high copy numbers of ARGs and MGEs in their genomes, were identified as critical STs involved in ARG dissemination within the species (Figure 6). The selected STs include ST357, ST234, ST316, ST773, ST697, ST175, ST111, ST277, ST621, ST167, ST233, ST654, ST1203, ST644, ST308, ST235, ST664, ST244, ST348, ST309, ST360, ST260, ST708, ST313, ST267, ST270, ST463, ST1182, ST1076, ST253, ST298, ST446, ST823, ST2211, ST1418, ST485, ST262, ST412, ST241, ST1976, ST1971, ST17, ST532, ST589, and ST549. The genome database survey provided a more expanded and consolidated list of STs involved in ARG transmission, compared to the literature analysis (Figure 3D and Figure 6). For example, 22 STs (ST234, ST316, ST621, ST644, ST348, ST260, ST708, ST267, ST270, ST1182, ST1076, ST2211, ST1418, ST485, ST262, ST412, ST241, ST1976, ST1971, ST17, ST532, and ST589) were exclusively detected through the genome database survey. On the contrary, ST463 which was frequently found in the literature analysis but not perceived as a high-risk clone, was derived as a critical ARG carrier ST by the genome database survey (Figure 3D and Figure 6). These results emphasize the reliability of the genome database survey to comprehend the relation between ST and ARG dissemination.

#### 3.2.4. Mobile ARGs

To prioritize ARGs that were mobile within the species, Pearson correlation analysis was employed while assessing the correlation between the copy numbers of the 382 acquired ARGs and integrases or transposases for each genome. ARGs that displayed a positive correlation (*r* value > 0.1) with statistical significance (*p*-value < 0.001) for integrases or transposases were considered mobile ARGs (Figure 7A). For plasmids, the copy number of ARGs in the plasmid contig was used. Among the 382 acquired ARGs, only 49 exhibited a correlation with at least one of integrase, transposase, or plasmid, thus signifying that these are mobile ARGs that should be given priority in surveillance efforts (Figure 7A). These mobile ARGs are primarily associated with human isolates but are also relatively prevalent in animal and environmental isolates, thus suggesting the need for monitoring from a One Health perspective (Figure 7B). Furthermore, these ARGs were distributed across numerous STs, thus endorsing their circulation within the species (Figure 7C). Aminoglycoside and β-lactam resistance genes were the most frequent types of mobile ARGs (Figure 7D). In terms of the number of kinds of ARGs, the proportion of mobile ARGs to total acquired ARGs was not significant (42 mobile ARGs out of 382 acquired ARGs). However, for β-lactam and aminoglycoside resistance genes, the total amounts of ARGs detected in the genomes for mobile ARGs accounted for more than half of those of the total acquired ARGs (Figure 7E). Notably, for sulfonamide, phenicol, and quinolone resistance genes, their abundance proportion relative to the acquired ARGs accounted for nearly the entire abundance, thus highlighting their significance as mobile ARGs (Figure 7E).

Given the reliability of the genome database survey, 49 ARGs can be directly postulated as mobile ARGs to be managed. However, it is crucial to also take account ARGs frequently encountered in the literature analysis, as their prevalence reflects research and clinical concerns on these ARGs (Appendix A). Red ARGs presented in Figure 7 represent ARGs that were also identified as mobile ARGs in our literature analysis, which exhibited a strong correlation with MGEs in the genome database survey. A total of 25 ARGs were revealed as mobile ARGs in both approaches; those should be prioritized as critical mobile ARGs. Remarkably, the other 24 ARGs, exclusively detected through the genome database survey, are also considered important mobile ARGs. For instance*, aph(3″)-Ib*, *aph(6)-Id*, and *aph(3′)-Via*, were identified as mobile ARGs solely through the genome database survey, but they exhibited robust associations with MGEs and a high prevalence in genomes, thus indicating that the genome database provided extensive profiles on mobile ARGs compared to the literature analysis (Figure 7A).

Regarding the 25 critical mobile ARGs, these ARGs exhibited frequent associations with MGEs in the literature analysis and displayed strong correlations with MGEs in the genome database survey. They included seven aminoglycoside resistance genes (*aac(6′)-Ib, ant(3″)-IIa, ant(2″)-Ia, aadA2, aac(6′)-Il, aac(3)-IIa,* and *aadA13*), ten β-lactam resistance genes (*bla*_VIM-1_, *bla*_OXA-1_, *bla*_OXA-2_, *bla*_OXA-10_, *bla*_KPC-2_, *bla*_GES-1_, *bla*_NDM-1_, *bla*_VEB-1_, *bla*_IMP-1_, and *bla*_IMP-9_), one MLSB resistance gene (*mphE*), one sulfonamide gene (*sul1*), three phenicol resistance genes (*floR*, *cmlA1*, and *catB3*), one quinolone resistance gene (*qnrVC1*), and two MDR efflux pump genes (*qacEΔ1* and *qacE*) (Appendix A).

## 4. Discussion

*P. aeruginosa* is a significant concern in the context of nosocomial infections, and the emergence and dissemination of antibiotic resistance within this species is of paramount importance. Concerning the clinical significance of *P. aeruginosa*, comparative genomics methodologies have been employed to elucidate its virulence and antibiotic resistance within the context of adaptation and evolution [44,45]. In general, the comparison of whole genome sequences of clinical isolates and their related strains available in the public database was executed to comprehend their phylogeny, STs, and ARG repertoires [46,47]. Additionally, genome-based investigations have been implicated in the prediction of antibiotic resistance of *P. aeruginosa* [30,48]. However, the systematic prioritization of critical factors for understanding ARGs, specifically through the utilization of comparative genomics encompassing all available genomes in the database which could offer comprehensive profiles of antibiotic resistance in the species, is scarce.

In this study, we assessed important factors, including mobile ARGs, major STs carrying mobile ARGs, critical MGEs involved in the dissemination of mobile ARGs, and the origins of mobile ARGs, using two distinct approaches that included a systematic literature analysis and a genome database survey. Appendix A summarizes the mobile ARGs, their major ST carriers, MGEs responsible for their dissemination, and identifies their One Health origins. Both approaches yielded consistent results; nevertheless, it is noteworthy that the genome database survey provided more extensive profiles of the critical factors in *P. aeruginosa*, whereas the literature analysis offered critical factors from the perspective of research and clinical concerns. Numerous studies have concentrated on clarifying high-risk STs and their ARG repertoires in association with MGEs [6,23,49].
Among mobile ARGs, aminoglycoside resistance and β-lactamase genes accounted for more than 60% of the list, coincided with the recent report highlighting the prevalent horizontal gene transfer of aminoglycoside and β-lactam resistance genes in *P. aeruginosa* [50]. The frequent occurrence and dissemination of clinically important β-lactamases, including carbapenemases and extended-spectrum β-lactamases (ESBL) in *P. aeruginosa* have been reported, which is consistent with our study’s list of mobile ARGs including carbapenemase genes (*bla*_NDM-1_, *bla*_KPC-2_, *bla*_VIM-1_, *bla*_IMP-1_, and *bla*_IMP-9_) and ESBL genes (*bla*_GES-1_, *bla*_VEB-1_, and *bla*_OXA-2_) [41,51].
The critical STs and mobile ARGs proposed in this study coincide with the findings from prior studies. More significantly, the current study offers an expanded list of critical STs and mobile ARGs using systematic analyses. Profiling and correlation analyses based on the comparative genomics of *P. aeruginosa* unveiled understated and potentially significant STs and mobile ARGs, emphasizing their importance for surveillance in clinical settings.

## 5. Conclusions

According to both approaches, 25 critical mobile ARGs (or 49 mobile ARGs based on the genome database survey), 45 critical STs, and associated MGEs responsible for ARG dissemination within the species, were elucidated as critical factors (Figure 3, Figure 7, and Appendix A). Additionally, animal and environmental isolates were identified as mobile ARG carriers (and also human isolates), thus emphasizing the need for monitoring and management from a One Health perspective (Figure 4, Figure 7 and Appendix A). Surveillance and management of these prioritized factors across One Health sectors are essential to mitigate the emergence of multidrug-resistant (MDR) and extensively resistant (XDR) *P. aeruginosa* in clinical settings.

## Figures and Tables

**Figure 1 ijms-24-15209-f001:**
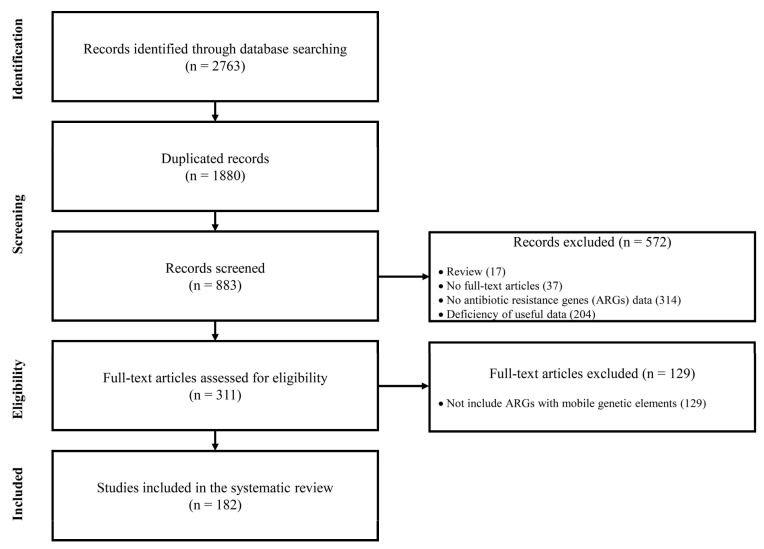
PRISMA (Preferred Reporting Items for Systematic Review and Meta-Analysis) flow diagram used for selection of studies regarding ARGs and their linkages to MGEs.

**Figure 2 ijms-24-15209-f002:**
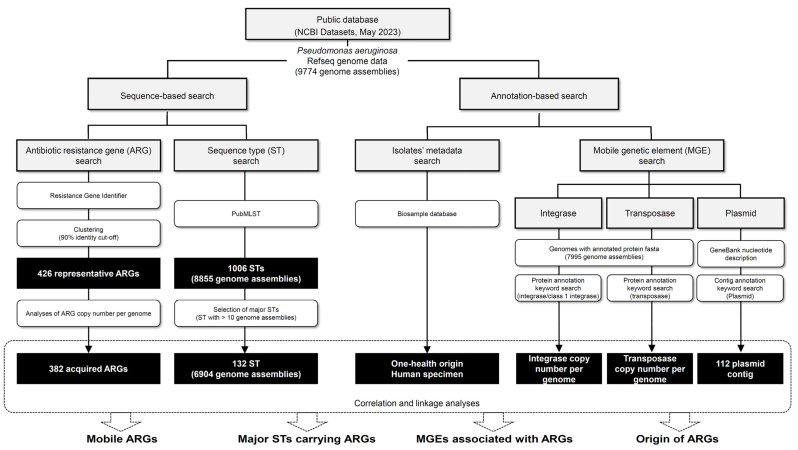
Flow diagram for a genome database survey for ARG, ST, MGE, and origin analyses.

**Figure 3 ijms-24-15209-f003:**
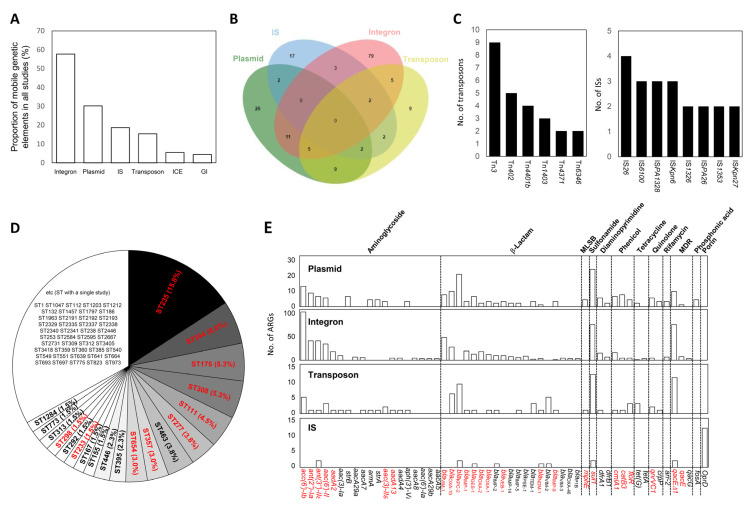
Characteristics and linkages of ARGs and MGEs identified by a systematic literature analysis. (**A**) Proportion of six types of MGEs; (**B**) Venn diagram indicating co-existence of MGEs [35]; (**C**) the number of types of transposons and ISs; (**D**) proportion of STs. Red letters indicate the top 10 high-risk clones; (**E**) the number of cases of ARGs observed in plasmids, integrons, transposons, and ISs. ARGs exhibiting the correlation with MGEs (more than three cases) are displayed. Red ARGs indicate mobile ARGs identified by a genome database survey (see the Figure in Section 3.2.4).

**Figure 4 ijms-24-15209-f004:**
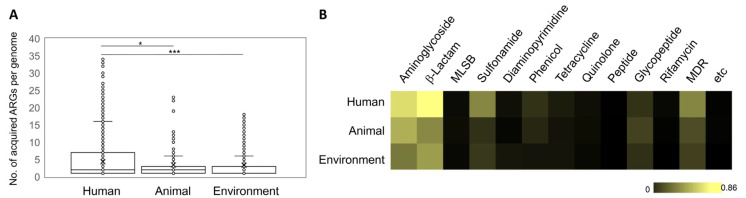
Abundance of acquired ARGs in One Health sectors. (**A**) the number of acquired ARGs in isolates from One Health sectors (*, *p* < 0.05; ***, *p* < 0.001); (**B**) heatmap displaying ARG abundance grouped into antibiotic classes in the One Health sector.

**Figure 5 ijms-24-15209-f005:**
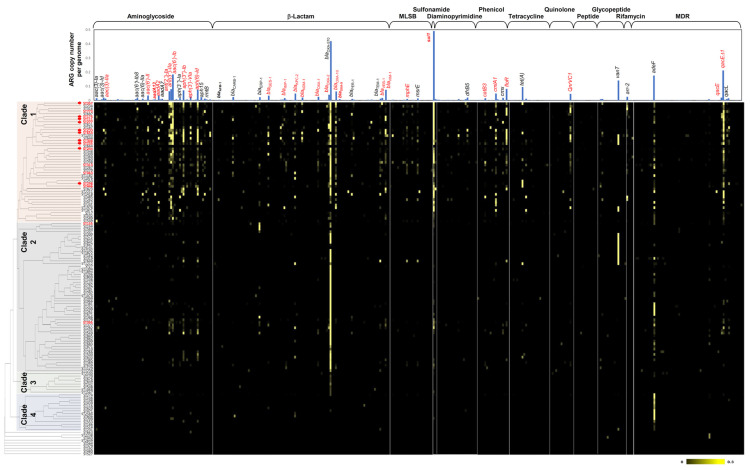
Abundance profiles of 382 acquired ARGs according to STs. The densities of the 382 acquired ARGs according to the STs are displayed as a heatmap. The overall density of each ARG in all genomes is indicated at the top as bar graphs. Left-axis clustering and distinct clades were obtained by the Bray–Curtis dissimilarity matrix of ARG repertoires of all genomes belonging to each ST. Red-colored ARG names indicate the coincidence of the prevalent ARGs in both systematic literature analysis and genome database survey. Major STs in the literature analysis are displayed in red (see Figure 3D) and the top 10 high-risk clones were marked with red arrows.

**Figure 6 ijms-24-15209-f006:**
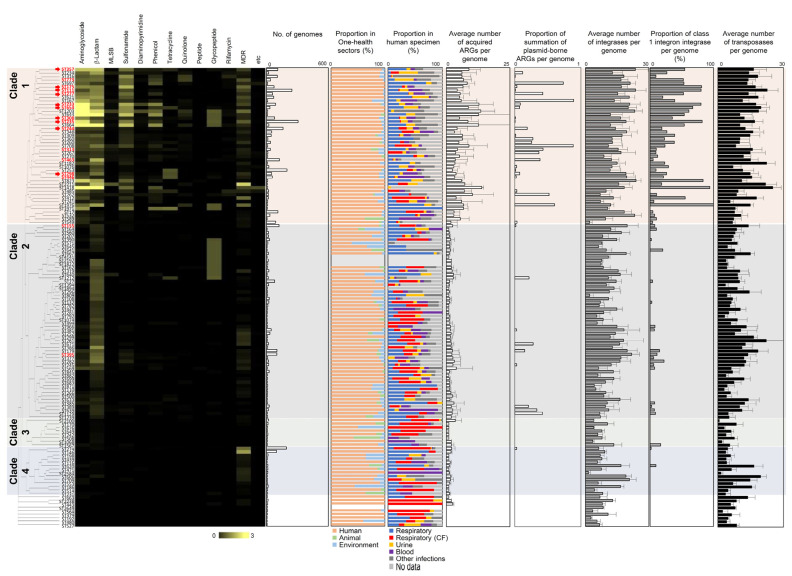
Association of MGEs and origins with profiles of ARGs in the STs. ARG abundances were grouped into antibiotic classes according to STs and are displayed as a heatmap. One Health origins and human specimens were presented as proportions. The proportion of plasmid-borne ARGs per genome was used to represent the occurrence of ARGs in plasmid among analyzed genomes. The average number of integrases and transposases per genome were used. For class 1 integrons, the proportion of class 1 integron integrases per genome was used. Major STs in the literature analysis are displayed in red (Figure 3D) and the top 10 high-risk clones were marked with red arrows.

**Figure 7 ijms-24-15209-f007:**
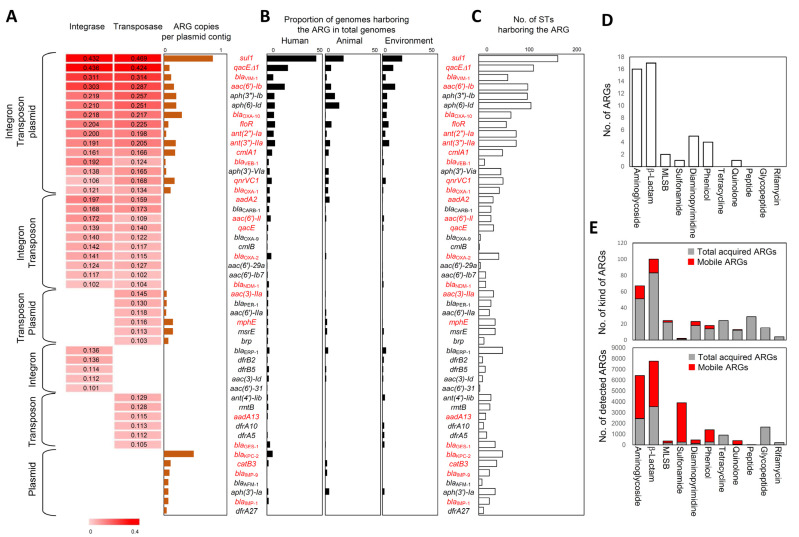
Mobile ARGs elucidated by MGE correlation and their significance. (**A**) ARGs exhibited associations with at least one MGE. ARGs exhibiting a statistically significant positive correlation (*r* > 0.1, *p* < 0.001) with integrase or transposase in Pearson correlation analysis were defined as mobile ARGs. For plasmids, ARG copy numbers per plasmid contig were used to define mobile ARGs; (**B**) One Health origin of mobile ARGs; (**C**) ST distribution of mobile ARGs; (**D**) types of mobile ARGs according to antibiotic class; (**E**) number and amount of mobile ARGs relative to those of acquired ARGs according to antibiotic class. Red ARGs indicate mobile ARGs identified by a literature analysis (see Figure 3E).

## Data Availability

Not applicable.

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
