# Peer review of "Prioritization of Critical Factors for Surveillance of the Dissemination of Antibiotic Resistance in Pseudomonas aeruginosa: A Systematic Review"

_ijms, 2023, doi:10.3390/ijms242015209_

Round 1
Reviewer 1 Report
Reviewer’s Comments:
The manuscript “Dissemination of Antibiotic Resistance Genes via Mobile Genetic Elements in Pseudomonas aeruginosa: Prioritizing Critical Factors” is a very interesting work. In this work, pseudomonas aeruginosa is the primary opportunistic human pathogens responsible for a range of acute and chronic infections, and it poses a significant threat to immunocompromised patients and is the leading cause of morbidity and mortality for nosocomial infections. Its high resistance to a diverse array of antimicrobial agents presents an urgent health concern. Among mechanisms contributing to resistance in P. aeruginosa, the horizontal acquisition of antibiotic resistance genes (ARGs) via mobile genetic elements (MGEs) has gained recognition as a substantial concern in clinical settings, thus indicating that a comprehensive understanding of ARG dissemination within the species is strongly required for surveillance. Here, two approaches, including a systematic literature analysis and a genome database survey, were employed to gain insights into ARG dissemination. The genome database enabled to scrutinize all available sequence information and various attributes of the isolates, thus providing an extensive understanding of ARG dissemination within P. aeruginosa. By integrating both approaches, with a primary focus on the genome database survey, mobile ARGs that were linked or correlated with MGEs, important sequence types (STs) carrying diverse ARGs, and MGEs responsible for ARG dissemination were identified as critical factors requiring strict surveillance. While I believe this topic is of great interest to our readers, I think it needs major revision before it is ready for publication. So, I recommend this manuscript for publication with major revisions.
1. In this manuscript, the authors did not explain the importance of Pseudomonas aeruginosa in the introduction part. The authors should explain the importance of Pseudomonas aeruginosa.
2) Title: The title of the manuscript is not impressive. It should be modified or rewritten it.
3) Correct the following statement “It is imperative to prioritize surveillance and management efforts for these critical factors across One Health sectors to mitigate the emergence of multidrug-resistant (MDR) and extensively resistant (XDR) P. aeruginosa in clinical settings”.
4) Keywords: The Pseudomonas aeruginosa is missing in the keywords. So, modify the keywords.
5) Introduction part is not impressive. The references cited are very old. So, Improve it with some latest literature like 10.1016/j.jallcom.2021.159013, 10.3390/molecules27217368
6) The authors should explain the following statement with recent references, “The density of a specific ARG within a specific ST was obtained by dividing the sum of the copy numbers of the specific ARG by the total number of genomes belonging to that specific ST”.
7) Add space between magnitude and unit. For example, in synthesis “21.96g” should be 21.96 g. Make the corrections throughout the manuscript regarding values and units.
8) The author should provide reason about this statement “The proportion of plasmid-borne ARGs per genome was used for plasmids”.
9. Comparison of the present results with other similar findings in the literature should be discussed in more detail. This is necessary in order to place this work together with other work in the field and to give more credibility to the present results.
10) Conclusion part is very long. Make it brief and improve by adding the results of your studies.
11) There are many grammatic mistakes. Improve the English grammar of the manuscript.
Minor editing of English language required
Author Response
October 12, 2023
Dear Reviewer 1, International Journal of Molecular Sciences (IJMS):
I appreciate the comments that you have made regarding our manuscript (Manuscript ID: ijms-2644526). I have carefully read your comments and the manuscript has been rewritten in response to these comments as detailed below.
The amended parts were represented and highlighted in yellow color in the revised manuscript (ijms-2644526-R1.docx).
I. Responses to comments of Reviewer 1
1. Comment: In this manuscript, the authors did not explain the importance of Pseudomonas aeruginosa in the introduction part. The authors should explain the importance of Pseudomonas aeruginosa.
Response: The parts “The bacterium’s versatile metabolic ability enables it to outcompete other bacteria in nutrient-limited conditions, suggesting its adaptability to hospital environments, leading in nosocomial infections [4]. The prevalence of P. aeruginosa in healthcare-associated infections underscores the significance of the bacterium as an opportunistic pathogen [5]” were added to the revised manuscript. These sentences were cited by recent references as follows.
- Cramer, N.; Klockgether, J.; Tümmler, B. Microevolution of Pseudomonas aeruginosa in the airways of people with cystic fibrosis. Curr. Opin. Immunol. 2023, 102328.
- Reynolds, D.; Kollef, M. The epidemiology and pathogenesis and treatment of Pseudomonas aeruginosa infections: an update. Drugs 2021, 81, 2117-2131.
2. Comment: Title: The title of the manuscript is not impressive. It should be modified or rewritten it.
Response: According to your recommendation, I changed the title in the revised manuscript as follows:
The title was changed to “Prioritization of Critical Factors for Surveillance of the Dissemination of Antibiotic Resistance in Pseudomonas aeruginosa”.
3. Comment: Correct the following statement “It is imperative to prioritize surveillance and management efforts for these critical factors across One Health sectors to mitigate the emergence of multidrug-resistant (MDR) and extensively resistant (XDR) P. aeruginosa in clinical settings”.
Response: The sentence was changed to “Surveillance and management of these prioritized factors across One Health sectors are essential to mitigate the emergence of multidrug-resistant (MDR) and extensively resistant (XDR) P. aeruginosa in clinical settings.” in the revised manuscript. The identical sentence in the conclusions section was similarly corrected as indicated.
4. Comment: Keywords: The Pseudomonas aeruginosa is missing in the keywords. So, modify the keywords.
Response: The keyword was added to the revised manuscript.
5. Comment: Introduction part is not impressive. The references cited are very old. So, Improve it with some latest literature like 10.1016/j.jallcom.2021.159013, 10.3390/molecules27217368.
Response: According to the DOI numbers, the recommended references are “Magnesium doped mesoporous bioactive glass nanoparticles: A promising material for apatite formation and mitomycin c delivery to the MG-63 cancer cells” (10.1016/j.jallcom.2021.159013) and “Synthesis, In Vitro Biological Evaluation and In Silico Molecular Docking Studies of Indole Based Thiadiazole Derivatives as Dual Inhibitor of Acetylcholinesterase and Butyrylchloinesterase” (10.3390/molecules27217368). After a careful examination of those articles, I acknowledge that certain contents within those articles could improve the quality of my manuscript. Notably, I observed that those articles did not encompass findings related to Pseudomonas aeruginosa or antibiotic resistance. In accordance with the insightful comments from the reviewer, I concur with the recommendation to incorporate the latest articles to make the manuscript impressive. Therefore, I improved the manuscript by incorporating the latest literatures, such as 10.1038/s41392-022-01056-1, 10.1016/j.coi.2023.102328, 10.1007/s40265-021-01635-6, and 10.1128/mSystems.00524-19. Among those, two are newly incorporated in the first paragraph of the introduction section (ref Nos. 4 and 5, related to response to comment 1) and the others replaced old ones (ref Nos. 3 and 16) in the introduction section.
- Qin, S.; Xiao, W.; Zhou, C.; Pu, Q.; Deng, X.; Lan, L.; Liang, H.; Song, X.; Wu, M. Pseudomonas aeruginosa: pathogenesis, virulence factors, antibiotic resistance, interaction with host, technology advances and emerging therapeutics. Signal Transduct. Target. Ther. 2022, 7, 199.
- Cramer, N.; Klockgether, J.; Tümmler, B. Microevolution of Pseudomonas aeruginosa in the airways of people with cystic fibrosis. Curr. Opin. Immunol. 2023, 102328.
- Reynolds, D.; Kollef, M. The epidemiology and pathogenesis and treatment of Pseudomonas aeruginosa infections: an update. Drugs 2021, 81, 2117-2131.
- Torrens, G.; Hernández, S.B.; Ayala, J.A.; Moya, B.; Juan, C.; Cava, F.; Oliver, A. Regulation of AmpC-driven β-lactam resistance in Pseudomonas aeruginosa: different pathways, different signaling. mSystems 2019, 4, e00524-19.
6. Comment: The authors should explain the following statement with recent references, “The density of a specific ARG within a specific ST was obtained by dividing the sum of the copy numbers of the specific ARG by the total number of genomes belonging to that specific ST”.
Response: The sentence was cited by a recent reference as follows:
- Lee, K.; Kim, D.W.; Cha, C.J. Overview of bioinformatic methods for analysis of antibiotic resistome from genome and metagenome data. J. Microbiol. 2021, 59, 270-280.
7. Comment: Add space between magnitude and unit. For example, in synthesis “21.96g” should be 21.96 g. Make the corrections throughout the manuscript regarding values and units.
Response: There seems to be an error in the editorial process (file conversion). Therefore, the part “21.96g” should be deleted. I deleted it in the revised manuscript.
8. Comment: The author should provide reason about this statement “The proportion of plasmid-borne ARGs per genome was used for plasmids”.
Response: As suggested, I modified the sentence to accurately convey the intended meaning as follows:
“The proportion of plasmid-borne ARGs per genome was used to represent the occurrence of ARGs in plasmid among analyzed genomes.”
9. Comment: Comparison of the present results with other similar findings in the literature should be discussed in more detail. This is necessary in order to place this work together with other work in the field and to give more credibility to the present results.
Response: As suggest, I have incorporated the discussion section to discuss our results by comparing them with precious studies. In addition, I added the relative references to the revised manuscript.
10. Comment: Conclusion part is very long. Make it brief and improve by adding the results of your studies.
Response: I have made it brief and improved by the results of our studies.
11. Comment: There are many grammatic mistakes. Improve the English grammar of the manuscript (Minor editing of English language required).
Response: Several grammatical errors were corrected in the revised manuscript.
I appreciate you for the time in reviewing this submission.
With best wishes,
Prof. Sang Hee Lee
_______________________________________________
National Leading Research Laboratory of Drug Resistance Proteomics
Department of Biological Sciences
Myongji University
116 Myongjiro, Yongin
Gyeonggido, 17058, Republic of Korea
TEL: +82-31-330-6195
FAX: +82-31-335-8249
e-mail: sangheelee@mju.ac.kr
Reviewer 2 Report
In their review manuscript “Dissemination of Antibiotic Resistance Genes via Mobile Genetic Elements in Pseudomonas aeruginosa: Prioritizing Critical Factors” the authors provide a broad overview of antibiotic resistance genes (ARG) encompassed on mobile genetic elements (MGE) in Pseudomonas aeruginosa, an important opportunistic pathogen causing life-threatening nosocomial infections. Their comprehensive approach using dual strategy of systematic literature analysis and a genome database survey affords deeper insights into diversity, co-occurrence and distribution of MGEs involved in ARG spreading within the species than literature survey alone would provide. This enabled identification of a number of critical factors (ARGs, associated MGE, and sequence types of the stains that harbour them) that can be of high clinical relevance for the spread of resistance in P. aeruginosa. These factors are discussed in the context of One Health, taking into account the ubiquitous nature of P. aeruginosa, its presence in animals and in the environment, which can be a reservoir of resistance determinants that can spread to human pathogenic strains as well.
The manuscript is appropriately structured and written and is of interest to the wider scientific community.
The manuscript is well-written and requires only minor adjustments (i.e. some singular/plural checking) particularly in the Abstract.
Author Response
October 12, 2023
Dear Reviewer 2, International Journal of Molecular Sciences (IJMS):
I appreciate the comments that you have made regarding our manuscript (Manuscript ID: ijms-2644526). I have carefully read your comments and the manuscript has been rewritten in response to these comments as detailed below.
The amended parts were represented and highlighted in yellow color in the revised manuscript (ijms-2644526-R1.docx).
I. Responses to comments of Reviewer 2
1. Comment: The manuscript is well-written and requires only minor adjustments (i.e. some singular/plural checking) particularly in the Abstract.
Response: Several grammatical errors were corrected in the revised manuscript.
I appreciate you for the time in reviewing this submission.
With best wishes,
Prof. Sang Hee Lee
_______________________________________________
National Leading Research Laboratory of Drug Resistance Proteomics
Department of Biological Sciences
Myongji University
116 Myongjiro, Yongin
Gyeonggido, 17058, Republic of Korea
TEL: +82-31-330-6195
FAX: +82-31-335-8249
e-mail: sangheelee@mju.ac.kr